# Molecular Determinants of Brevetoxin Binding to Voltage-Gated Sodium Channels

**DOI:** 10.3390/toxins11090513

**Published:** 2019-09-03

**Authors:** Keiichi Konoki, Daniel G. Baden, Todd Scheuer, William A. Catterall

**Affiliations:** 1Department of Pharmacology, Box 357280, University of Washington, Seattle, WA 98195-7280, USA; 2Center for Marine Science, University of North Carolina, Wilmington, NC 28409, USA

**Keywords:** neurotoxic shellfish poisoning, voltage-gated sodium channels, binding assay

## Abstract

Brevetoxins are produced by dinoflagellates such as *Karenia brevis* in warm-water red tides and cause neurotoxic shellfish poisoning. They bind to voltage-gated sodium channels at neurotoxin receptor 5, making the channels more active by shifting the voltage-dependence of activation to more negative potentials and by slowing the inactivation process. Previous work using photoaffinity labeling identified binding to the IS6 and IVS5 transmembrane segments of the channel α subunit. We used alanine-scanning mutagenesis to identify molecular determinants for brevetoxin binding in these regions as well as adjacent regions IVS5-SS1 and IVS6. Most of the mutant channels containing single alanine substitutions expressed functional protein in tsA-201 cells and bound to the radioligand [42-^3^H]-PbTx3. Binding affinity for the great majority of mutant channels was indistinguishable from wild type. However, transmembrane segments IS6, IVS5 and IVS6 each contained 2 to 4 amino acid positions where alanine substitution resulted in a 2–3-fold reduction in brevetoxin affinity, and additional mutations caused a similar increase in brevetoxin affinity. These findings are consistent with a model in which brevetoxin binds to a protein cleft comprising transmembrane segments IS6, IVS5 and IVS6 and makes multiple distributed interactions with these α helices. Determination of brevetoxin affinity for Na_v_1.2, Na_v_1.4 and Na_v_1.5 channels showed that Na_v_1.5 channels had a characteristic 5-fold reduction in affinity for brevetoxin relative to the other channel isoforms, suggesting the interaction with sodium channels is specific despite the distributed binding determinants.

## 1. Introduction

After brevetoxins A (PbTx1) and B (PbTx2) were isolated from the dinoflagellate *Karenia brevis* (formerly named *Ptycodiscus brevis* or *Gymnodinium breve*) by Lin et al. [1] and Shimizu et al. [2] a series of cognates were purified, named PbTx-1 to 9 after the original species designation of the dinoflagellates [3]. Blooms of this harmful dinoflagellate in the Gulf of Mexico in 1996 and New Zealand in 1992 to 1993 caused massive kills of fish and mammals. The aerosol containing the toxin induces non-fatal effects on human health including skin irritation, non-productive cough, shortness of breath and tearing. Ingestion of shellfish exposed to the bloom results in neurotoxic shellfish poisoning (NSP) with gastrointestinal and neurological sequelae including peripheral and central nervous system injury in humans. Those biological effects were attributed to the action of PbTx-B1 to B4 [4,5,6] which are thought to be metabolic products of brevetoxin B in the shellfish body.

Structures of brevetoxins are characterized by a linear array of cyclic ethers with trans/syn-fused ether rings (Figure 1A). This general structure is shared with other polycyclic ether toxins such as maitotoxin [7], ciguatoxin [8], prymnesin [9], gambierol [10], gambieric acid [11], yessotoxin [12], brevenal [13] and bremisamide [14]. All of these toxins exhibit a diverse range of potent biological activities, which have motivated many scientists to investigate their molecular targets as well as the underlying mechanism of actions [15,16,17].

Brevetoxins act on voltage-gated sodium channels [15]. Brain sodium channels consists of an α subunit containing the ion-conducting pore and the gating machinery that allows the channel to open in response to voltage and two auxiliary β subunits, one of which is disulfide-linked to the α subunit [18]. The α subunit consists of four homologous transmembrane domains, each of which contains six transmembrane segments (S1–S6) and a pore-forming region between S5 and S6 termed SS1–SS2 (Figure 1B). The α subunit is characterized by more than six independent biochemically-defined neurotoxin receptor sites, 1 through 6. Brevetoxins act on the sodium channel by binding to neurotoxin receptor site 5. It results in persistent channel activation by producing a negative shift in the voltage dependence of activation and slowing the rate of the critical inactivation process [19,20]. In addition, at the single channel level, it produces sub-conductance states [21]. Binding of brevetoxin to site 5 allosterically affects the action of toxins binding at neurotoxin receptor site 2 such as batrachotoxin and veratridine, and of toxins binding at site 4, such as scorpion toxins [22]. However, neither site 2 nor site 4 toxins affect brevetoxin binding [3,22].

Ciguatoxins are responsible for ciguatera, a food poisoning prevailing in tropical and subtropical waters and affecting more than ten thousand people per year with significant morbidity, particularly in the atoll island countries of the Pacific basin [23]. The toxin binds competitively with brevetoxins at neurotoxin site 5 [24]. In contrast to brevetoxin, it shows strong acute toxicity to mice and cytotoxic activity. The binding affinity of ciguatoxin-1B to rat brain sodium channel is 50-fold stronger than that of PbTx1 [25]. Limited amounts of ciguatoxin had long prevented biological studies to elucidate the mode of actions for ciguatoxin, though total synthesis of ciguatoxins by Hirama’s group has enabled them to overcome the deficit and elucidate its mode of action in detail [26,27,28]. Nevertheless, the portion of the structure required for the enhanced binding to the voltage-sensitive sodium channels and the mechanism underlying its toxicity in vivo has not been determined. Studies of the highly related brevetoxins can provide considerable insight into the molecular actions of ciguatoxins. Both a series of natural products and synthetic derivatives, intermediates and truncated forms of brevetoxins have been tested and provide useful molecular information.

Photoaffinity labeling of sodium channels in rat brain synaptosomes and reconstituted purified sodium channels with brevetoxin has been used to identify the brevetoxin binding site [29]. Using proteolytic digestion by *N*-tosyl-l-phenylalanine chloromethyl ketone (TPCK) trypsin followed by immunoprecipitation of the resulting sodium channel peptides with site-directed anti-peptide antibodies has identified the IS6 and IVS5 segments as being photolabeled and as possible sites of brevetoxin binding [29].

Here we performed experiments to further narrow the brevetoxin binding site. First, we used alanine-scanning mutagenesis of transmembrane segments IS6 and IVS5 as well as nearby IVS5-SS1 and IVS6 segments to identify molecular determinants of brevetoxin binding. In addition, we compared binding to Na_v_1.2 with binding to sodium channel isoforms Na_v_1.4 and Na_v_1.5 to detect differences in binding affinity. These experiments suggest that brevetoxin has a distributed hydrophobic binding site and makes interactions with multiple sites on all channel regions tested.

## 2. Results

### 2.1. Alanine Scanning of Transmembrane Segments

TsA-201 cells were transiently transfected with cDNAs encoding wild-type (WT) or alanine mutant Na_v_1.2 α subunits. Membrane fractions were isolated and incubated for 3 h with various concentrations of [42-^3^H]-PbTx3, adequate time for the reaction to reach equilibrium. Saturation binding curves were well fit with a one-site binding model (Figure 2) to obtain K_d_ and B_max_. Using this model, the dissociation constant of the ligand from wild type Na_v_1.2 was 2.4 ± 0.2 nM (mean ± standard error of the mean (SEM, *n* = 30), comparable to K_d_ values obtained previously [3].

Since the sites photolabeled by brevetoxin binding were in transmembrane segments IS6 and IVS5 [30], alanines were substituted individually for each amino acid residue in both regions. Substitution of alanine was expected to reduce side chain interactions of the native residue, including electrostatic and hydrophobic interactions. Each mutant construct was transfected into tsA-201 cells and [42-^3^H]-PbTx3 binding was determined.

The results of these experiments for mutants in transmembrane segment IS6 are summarized in Figure 3A. Of the 23 positions with non-alanine native residues, 22 gave adequate protein expression for binding experiments. Na_v_1.2 L421A was not expressed in tsA-201 cells. Alanine mutants at most residues gave K_d_ values similar to that for wild type. Binding to Na_v_1.2 constructs with alanine substitutions at M402, L407 and V408 increased the K_d_ by 2–3-fold (Figure 3A); G412A, F414A and Y415A caused a 2–3-fold decrease in K_d_ (Figure 3A). Thus, substitution of individual amino acids with alanine in IS6 can increase or decrease binding affinity for brevetoxin by modest increments.

Transmembrane segment IVS5 is composed of amino acids from N1661 to V1685. Of the 23 positions with non-alanine native residues, all mutants in this segment were expressed as measurable proteins. Mutant N1662A protein was detected in immunoblot using anti-SP20 but did not bind to brevetoxin. In addition, no measurable current was observed using whole-cell patch-clamp recording of transfected cells. Mutants F1672A and I1673A were barely expressed, which suggests that these mutations render the channel nonfunctional. As for IS6, most alanine mutants of residues in IVS5 bound to brevetoxin with affinities similar to wild type (Figure 3B). However, I1663A, G1664A, L1665A and L1666A reduced affinity, giving K_d_ values 2–3-fold greater than WT. In contrast, L1669A, V1670A, G1678A and Y1684A increased affinity approximately 2-fold.

The peptide fragment photolabeled by brevetoxin included a portion of the IVS5-SS1 loop [29]. Therefore alanine-scanning mutagenesis was extended into IVS5-SS1 loop to F1697 (Figure 3C). As in the other regions, most substitutions had little effect on K_d_. However, E1688A and F1697A increased binding affinity of the ligand by approximately 2-fold.

Regions identified by photoaffinity labeling are likely to be near the ligand-binding site but may not indicate the binding site itself. The linker connecting the photoaffinity tag and the ligand must be short so that the target is accurately identified. However, it must be long enough so that the tag does not interfere with binding of the ligand to its target. In addition, photoactive species have a characteristic lifetime after activation by irradiation with UV. This lifetime has to be long enough to interact with target molecules but not so long as to yield nonspecific labeling of distant peptides. Due to these factors, the site of photolabeling in targets can be distant from critical binding determinants for the ligand, which may lead to failure to identify those determinants. An example is provided by the polypeptide α-scorpion toxin where some of the regions determined by photolabeling were not found to be critical for binding or function of the toxin [31,32]. Since only small effects of alanine substitution on brevetoxin binding were observed in IS6 and IVS5, we extended analysis to a region that lies between IS6 and IVS5 in the folded structure of the sodium channel, transmembrane segment IVS6 [33].

Alanine mutants in IVS6 were tested for brevetoxin binding (Figure 3D). Of the 24 mutant channels studied, S1758A decreased K_d_ 2-fold and F1756A, I1760A, I1761, Y1771A and I1772A each resulted in an increase in K_d_ for binding. Thus, like alanine substitutions in IS6 and IVS5, substitutions in IVS6 caused multiple small changes in toxin binding affinity.

### 2.2. Sodium Channel Isoform-Dependent Differences in Toxin Binding

Differences in affinity between sodium channel isoforms can provide important clues for identifying molecular determinants of binding. This strategy has been used successfully for identifying sodium channel binding determinants of α- and β-scorpion toxin [32,34]. To identify any such isoform-dependent differences in brevetoxin affinity, we examined binding to transiently expressed Na_v_1.4 and Na_v_1.5. Binding affinity of Na_v_1.4 channels for the toxin was similar to that of Na_v_1.2 (Figure 4A). However, the K_d_ for binding to Na_v_1.5 increased by approximately 5-fold to 12 ± 1.4 nM (mean ± SEM, *n* = 6). Na_v_1.4 has 1.8 ± 0.61 nM (mean ± SEM, *n* = 5) of K_d_. We further characterized the differences between brevetoxin binding to Na_v_1.2 and Na_v_1.5 by measuring [42-^3^H]-PbTx3 dissociation rates (Figure 4B). The rate constant for toxin dissociation from Na_v_1.2, k_d-1.2_, was 0.037 ± 0.010 min^−1^ whereas that for dissociation from Na_v_1.5, k_d-1.5_, was 0.13 ± 0.02 min^−1^, about 4-fold faster. The inverse ratio of these dissociation rates, k_d-1.5_/k_d-1.2_, is comparable to the ratio of dissociation constants for Na_v_1.2 and Na_v_1.5, K_d-1.2_/K_d-1.5_. This suggests that most of the difference in the affinity of Na_v_1.2 and Na_v_1.5 for brevetoxin can be attributed to the difference in dissociation rates.

## 3. Discussion

Photolabeling identified transmembrane segments IS6 and IVS5 as being near the brevetoxin binding site [29]. We explored these regions as well as an adjacent transmembrane α helix, IVS6 for involvement in brevetoxin binding. In addition, we examined the C-terminal extracellular extension of IVS5, IVS5-SS1, for effects on brevetoxin binding. These experiments identified residues in each segment that reduced binding affinity by 2–3-fold. They also identified residues that increased by a similar factor. In addition to identifying these residues affecting binding, comparison of binding to sodium channel isoforms Na_v_1.2, Na_v_1.4 and Na_v_1.5 identified a 5-fold reduction in affinity for binding to the Na_v_1.5. This represents the simultaneous comparison of brain, muscle and cardiac sodium channel isoforms in terms of binding to brevetoxin [35,36] and is consistent with a finding that Na_v_1.5 exhibits a lower affinity than Na_v_1.4 [35].

Brevetoxin is about 30 Å in length and comprises conjugated ether rings (Figure 1A). The overall form of the molecule with hydrophilic groups on either end connected by the chain of hydrophobic ether rings suggests a transmembrane topology of the bound molecule [37,38]. Distributed ether oxygens are thought to make hydrogen bonds with nearby amino acids of the channel while hydrocarbons comprising most of the molecule might stabilize it in the membrane or body of the channel by hydrophobic interactions.

Well-defined molecular structures are available for the voltage-gated sodium channels [39,40]. These structures place the pore-lining S6 segments adjacent to each other. In the context of the sodium channel a brevetoxin molecule could be imagined intercalating between the two S6 segments and the transverse S5 segment. Recent structures of voltage-gated channels provide additional insight into other transmembrane helices that might contribute binding determinants to the brevetoxin-channel interaction [39,40]. One possible candidate is the S4 segment(s) of domain III or domain IV. Structurally the voltage sensor region of domain IV containing the S4 segment is near the pore region of domain I. Interactions with these segments would provide a molecular substrate for the effects of brevetoxin on voltage-dependent activation of the channel since the charges on the S4 segments are the voltage sensors for channel activation.

The pattern of effects of individual alanine mutations on brevetoxin affinity gives some insight into the nature of brevetoxin binding. The extended form of the molecule suggests that it lies at the interface between α helices and fits quite tightly there [37,41]. In this configuration, the native amino acids in the α helices would make multiple interactions with the brevetoxin molecule, some attractive and some repulsive. In concert with this hypothesis, our experiments revealed multiple positions where introduction of an alanine produced small effects on brevetoxin affinity and these effects were bi-directional, some causing increases in affinity and some causing decreases. At positions where alanine caused decreased affinity and where it replaced a bulkier residue, the loss of the amino acid side chain might have removed an interaction with the molecule. At other positions such as 1664, 1690 and 1752 where alanine replaces a glycine, substituting the smaller amino acid might have made the local “fit” of the molecule less good. Conversely, at positions where replacement of the native amino acid with alanine resulted in an increase in affinity, the change might have provided a favorable “fit” with the backbone of the brevetoxin molecule. In this scheme there will be interactions with individual residues but much of the binding affinity may be distributed over a large number of residues. In addition, the extremely hydrophobic nature of the molecule suggests that a large part of its binding energy derives from that hydrophobicity and the stability derived from being in a hydrophobic environment [42].

## 4. Materials and Methods 

### 4.1. Materials

PbTx-3 and [42-^3^H]-PbTx3 (16 Ci/mmol) were prepared as described [3]. Labeling of brevetoxin on position 42 has been shown to have little effect on the affinity of PbTx for the sodium channel [29]. PbTx-1 was purchased from Calbiochem (Darmstadt, Germany).

### 4.2. Molecular Biology

Construction of alanine mutants in IS6 and IVS6 segments was previously described [43,44]. Each of these mutant constructs was subcloned into pCDM8. To prepare mutants in IVS5 a Blp I to Cla I fragment of pCDM8 Na_v_1.2, which encodes part of domain III, domain IV and *C*-terminus of Na_v_1.2, was subcloned into pBSII KS(+). With this shuttle vector as a template, a two-step PCR mutagenesis protocol using two mutagenic primers and two restriction site primers was used to introduce alanine mutants. The mutagenic fragment and shuttle vector were digested with BamH I restriction endonuclease. The mutagenic fragment was then subcloned into the shuttle vector via the restriction sites. Mutations were confirmed by restriction mapping and DNA sequence analysis. The Blp I to Cla I restriction fragment was then excised from pBSII and subcloned into pCDM8 Na_v_1.2.

### 4.3. Transfection of Na_v_1.2 into tsA-201 Cells

TsA-201 cells were maintained at 37 °C in 10% CO_2_/air in Dulbecco′s modified eagle’s medium (DMEM)/F12 medium (Invitrogen, Carlsbad, CA, USA) supplemented with 10% fetal bovine serum, 20 µg/mL penicillin and 10 µg/mL streptomycin. A 150 mm plate of tsA-201 cells at 80% confluence was split into 2 new plates. After 8–10 h when the cells had adhered and regained their normal shape, a 1:10 volume of DNA-calcium phosphate complex was added in a dropwise fashion [45], using 60 µg/plate of pCDM8 Na_v_1.2 and 3 µg/plate of pCDNA3.1 β1. The plates were maintained in a 3% CO_2_/air incubator for 12 h. The medium was then replaced, and the cells were maintained for an additional 24 h in a 5% CO_2_/air incubator before use in binding assays.

A high level of expression is critical for binding studies like those reported here and is extremely sensitive to the pH of the transfection buffer [30]. We typically prepared *N*,*N*-bis(2-hydroxyethyl)-2-aminoethanesulfonic acid (BES) solutions with pH varying from 6.85 to 7.05 and compared them in transfection experiments followed by binding of [42-^3^H]-PbTx3 as described below. The BES solution with the pH that gave the greatest B_max_ for [42-^3^H]-PbTx3 was used.

### 4.4. Preparation of Membrane Fractions

Unless otherwise noted, all procedures were performed at 0–4 °C. The medium was aspirated, and the cells were washed with 10 mL of phosphate-buffered saline (PBS). Then, 3 mL of PBS was added to the dish before harvesting the cells with a cell scraper. After centrifugation at 700 *g* for 5 min, the cells were resuspended in 50 mM Tris/HCl (pH 8.0) containing a mixture of protease inhibitors (1 µM pepstatin A, 10 µM benzamidine, 1 µM leupeptin, 35 µg/mL PMSF and 1 µg/mL aprotinin) and passed 4 times through a needle (25G 1-1/2”, Becton Dickinson, Franklin Lakes, NJ, USA). The lysate was centrifuged at 700 *g* for 5 min, and the resulting supernatant was centrifuged at 21,000 *g* for 30 min. The pellet containing the membrane fraction was resuspended in 1.0 mL of binding buffer containing 50 mM Hepes/Tris (pH 7.5), 130 mM choline chloride, 5.4 mM KCl, 0.5 mM MgSO_4_, 5.5 mM glucose and 0.01% polyoxyethylene 10 tridecyl ether, and stored at –80 °C until use. Protein concentrations were measured using a BCA Protein Assay Kit (Pierce, Waltham, MA, USA).

### 4.5. In Vitro Binding

Next, 250 µg of the membrane fraction suspended in 500 µL of the binding buffer was incubated with the appropriate concentration (0.5–10 nM) of [42-^3^H]-PbTx3 for 3 h. The mixture was diluted with 2.5 ml of washing buffer containing 163 mM choline chloride, 5 mM Hepes (pH 7.5), 1.8 mM CaCl_2_, 0.8 mM MgSO_4_ and 100 µg/mL bovine serum albumin (BSA) and poured onto a glass fiber filter GF/C (φ25 mm, Whatman, Maidstone, UK) mounted on a vacuum manifold (1225 Sampling Manifold, Millipore, Burlington, MA, USA), which was connected to a vacuum aspirator. The glass fiber filter was washed 3 times with this buffer, mixed with 5 mL of EcoLume Liquid Scintillation Cocktail (MP Biomedicals, Irvine, CA, USA) and incubated overnight before counting radioactivity with Multi-purpose Scintillation Counter LS6500 (Beckman Coulter, Inc., Brea, CA, USA). Nonspecific binding was measured in the presence of 600 nM cold PbTx1 or PbTx3. Saturation binding curves were nonlinearly fitted to an equation for 1:1 receptor-ligand interactions using GraphPad Prism 2.0 (GraphPad Software, San Diego, CA, USA) to estimate K_d_ and B_max_. Experiments were done in duplicate and the reported number of experiments represents repetitions on different days.

Ligand dissociation was measured as follows: first, 250 µg of the membrane fractions in 500 µL of binding buffer was incubated with 0.5–10 nM of [42-^3^H]-PbTx3 at 4 °C for 3 h as described above. Next, 2.5 mL of ice-cold buffer containing 600 nM PbTx1 was added. After periods of 2, 4, 8, 16, and 32 min, the remaining radioactivity bound to the membranes was measured as described above.

A few mutant channels (e.g., L1662A) were detected by immunoblot with anti-SP20 which recognizes the Na_v_1.2 α subunit but did not bind to [42-^3^H]-PbTx3. In these cases, whole-cell voltage clamp recording was performed in cells expressing these channels to detect surface expression of functional channels. We were concerned that misfolding of particular mutant channels might have resulted in inconsistent binding, perhaps because the structure was sensitive to purification of membrane fraction. To ameliorate this several approaches were tried without success. Hypotonic, isotonic or hypertonic buffer were used when the harvested cells were homogenized but did not produce significant differences in the results. We also tried detaching transfected cells from the dishes with EDTA-containing PBS and using the cells directly for binding studies without preparing membrane fragments, but this method failed due to elevated nonspecific binding of [42-^3^H]-PbTx3.

## Figures and Tables

**Figure 1 toxins-11-00513-f001:**
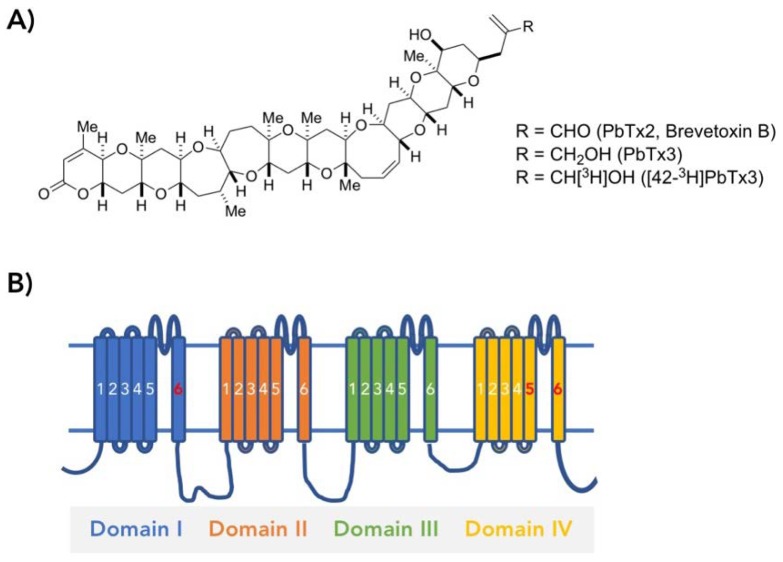
(**A**) Structures of brevetoxin B (PbTx2), PbTx3 and [42-^3^H]PbTx3; (**B**) Schematic figure of the sodium channel α subunit. The α subunit consists of four homologous transmembrane domains, each of which contains six transmembrane segments (S1–S6). A series of alanine mutants in three segments, including IS6, IVS5 and IVS6, was prepared for the binding assays.

**Figure 2 toxins-11-00513-f002:**
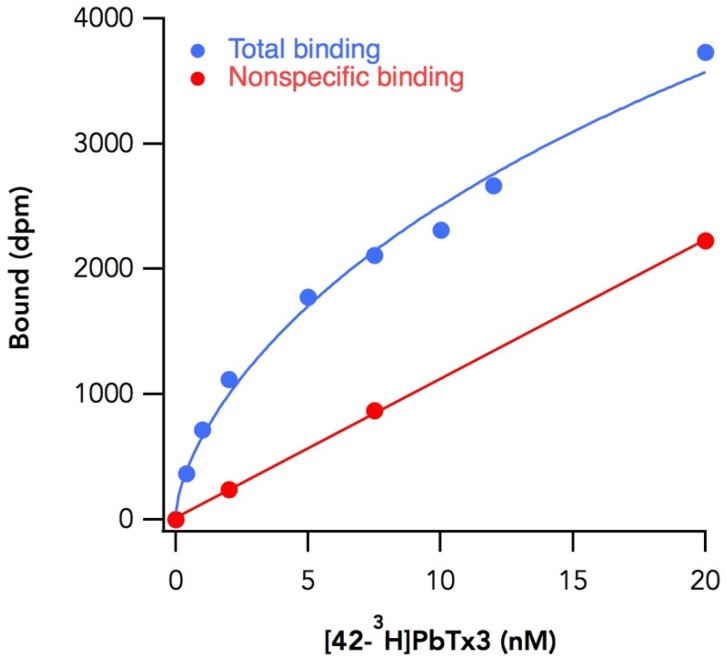
Binding of [42-^3^H]-PbTx3 to alanine mutants for IS6. A total of 250 µg of membrane fractions were incubated with eight different concentrations of the ligand (0.5–10 nM) in the presence (red) or absence (blue) of non-labeled PbTx3. For a particular experiment, each concentration was studied in duplicate. Experiments were repeated at least twice for each mutant. Saturation binding curves were fitted nonlinearly to a 1:1 ligand–receptor interaction to calculate K_d_ and B_max_.

**Figure 3 toxins-11-00513-f003:**
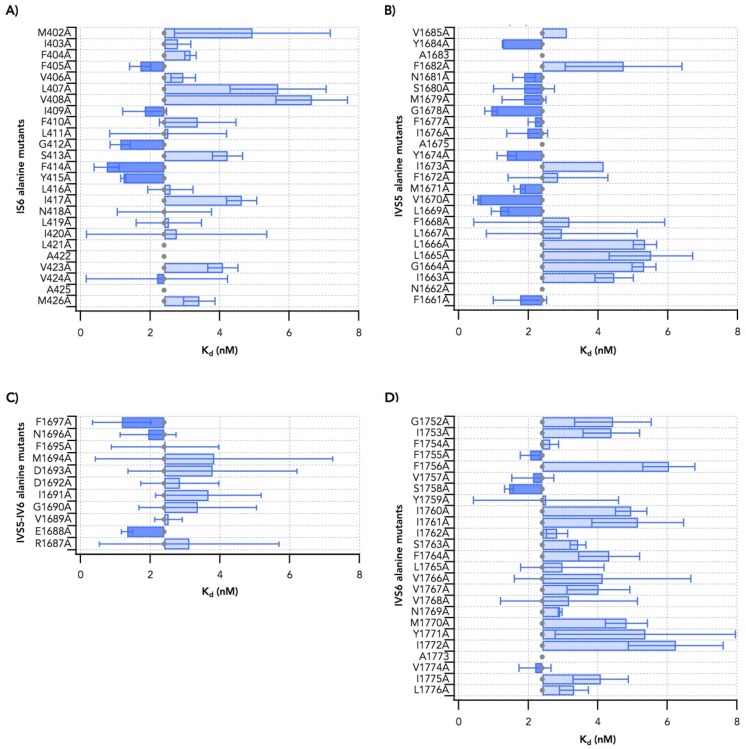
Binding of [42-^3^H]-PbTx3 to alanine mutants for (**A**) IS6, (**B**) IVS5, (**C**) IVS5-SS1 loop, and (**D**) IVS6.

**Figure 4 toxins-11-00513-f004:**
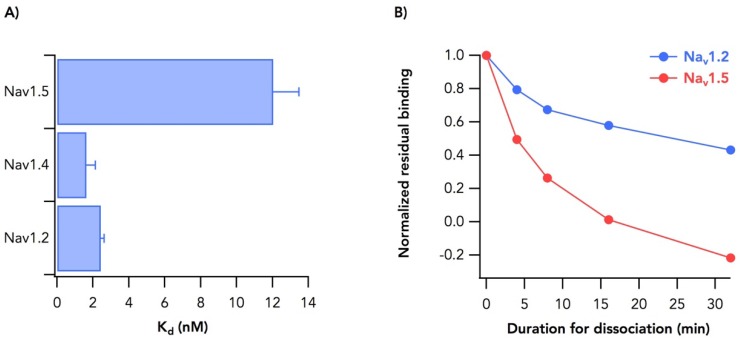
Binding of [42-^3^H]-PbTx3 to different channel subtypes. (**A**) Dissociation constants of [42-^3^H]-PbTx3 against Na_v_1.2, 1.4 and 1.5. (**B**) Dissociation kinetics for Na_v_1.2 (blue) and 1.5 (red).

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
