# Peer review of "Molecular Determinants of Brevetoxin Binding to Voltage-Gated Sodium Channels"

_toxins, 2019, doi:10.3390/toxins11090513_

Round 1

Reviewer 1 Report

L61, L137: (alpha?)-scorpion toxin, please insert ().

L205: "fit' -> "fit"

L210~274:Heading and subheadings should be changed as follows: 

5.Materials and Methods -> 4. Materials and Methods,

4.1. Materials, 4.2. Molecular Biology, ----

L345: 28. 2Yamaoka -> Yamaoka

Author Response

L61 (69), L137 (147): (alpha?)-scorpion toxin, please insert () : Thank you very much. The system error occurred. Alpha and beta in symbol format were not displayed correctly. We all corrected.

L205 (216): "fit' -> “fit" Thank you very much. We corrected it.

L210 (221)~274 (285):Heading and subheadings should be changed as follows:  Thank you very much. We corrected it.

L345 (366): 28. 2Yamaoka -> Yamaoka Thank you very much. We corrected it.

Reviewer 2 Report

In this study alanine-scanning mutagenesis was used to identify residues whose substitutions affect binding of brevetoxin to three sodium channels. The effects of the substitutions are mild (2-3 fold reduction of affinity) and it is unclear whether mutated residues form direct contacts with the toxin or mutations allosterically affect binding of brevetoxin. Nevertheless, the results are consistent with earlier studies and obtained data may be useful for future elaboration of 3D models  of brevetoxin-bound ion channels. Also interesting is the finding that the Nav1.5 channel is less sensitive to brevetoxin than Nav1.2 and Nav1.4.

Minor

Line 49.  "Brain sodium channels are heterotrimeric proteins ..."  All Nav1.x channels have the same subunit composition

Line 61: "-scorpion toxins"  ?

Line 98:  "cold ligand"   ?

Figure 3 A and D:  Wrong axis captions

Figure 3 B - D:  Why bars are shaded differently (cf. Fig. 3wA)

Line 137: "-scorpion toxin"  ?

Line 140: "... that is predicted... "  Cryo-EM structures do show disposition of various segments

Figure 4B:  Wrong axes names

Line 152:  "- and -scorpion toxin"  ??

Line 155:   "was increased"   Change to  "increased"

Line 176:  "transmembrane distribution" Change to "transmembrane topology"

Lines 179-181. The two sentences are speculations.  Please delete.

Lines 184-185.  Please remove  "with S5 segment passing diagonally across them"

Line 188.  "S4 segment(s) of domain I or domain IV. "  IS4 is far from the mutated residues.  IIIS4 and IVS4 are rather close.

Line 191.  "are thought"  S6s are voltage sensors.

Lines 193-197.  Speculations. Conflict with statement at lines 179-180.  Please remove or change.

Line 196: "positive and negative interactions" Attractive and repulsive interactions   

Author Response

Line 49.  "Brain sodium channels are heterotrimeric proteins ..."  All Nav1.x channels have the same subunit composition. Thank you very much. We changed it to “Brain sodium channels consisits of an …” Line 61 (71): "-scorpion toxins”? Thank you very much. The system error occurred. Alpha and beta in symbol format were not displayed correctly. We all corrected. Line 98:  "cold ligand"? We changed to “non-labelled PbTx3” Figure 3 A and D:  Wrong axis captions Thank you very much for your comment. We corrected. Figure 3 B - D:  Why bars are shaded differently (cf. Fig. 3A). We shaded the bars with the same domain color as was used Figure 1. As Reviewer 2 did not notice it, however, the color might have not caused a significant difference. We decided to use only one color. Please see the revised Fig. 3. Line 137 (147): "-scorpion toxin”? Thank you very much. The system error occurred. Alpha and beta in symbol format were not displayed correctly. We all corrected. Line 140: "... that is predicted... "  Cryo-EM structures do show disposition of various segments. Thank you very much for you comment. We changed from “that is predicted to lie” to “that lies” Figure 4B:  Wrong axes names. Thank you very much for your comment. We corrected. Line 152:  "- and -scorpion toxin" ?? Thank you very much. The system error occurred. Alpha and beta in symbol format were not displayed correctly. We corrected. Line 155:   "was increased"   Change to  “increased”. We corrected it. Line 176:  "transmembrane distribution" Change to "transmembrane topology” Thank you very much. We corrected it. Lines 179-181. The two sentences are speculations.  Please delete. Thank you very much. We deleted it. Lines 184-185.  Please remove  "with S5 segment passing diagonally across them” Thank you very much. We deleted it. Line 188.  "S4 segment(s) of domain I or domain IV. "  IS4 is far from the mutated residues.  IIIS4 and IVS4 are rather close. Thank you very much for your comment. Thank you very much. We changed to "S4 segment(s) of domain IIII or domain IV ". Line 191.  "are thought"  S6s are voltage sensors. Thank you very much. We removed “thought to be” Lines 193-197.  Speculations. Conflict with statement at lines 179-180.  Please remove or change. Thank you very much for your comment. We remove the lines 179-180 and leave Lines 193-197 as it was. Line 196: "positive and negative interactions" Attractive and repulsive interactions . Thank you very much. We correct it as “attractive and repulsive”.